# DinoSR: Self-Distillation and Online Clustering for Self-supervised Speech Representation Learning

**Alexander H. Liu**[†]    **Heng-Jui Chang**[†]    **Michael Auli**[‡*]    **Wei-Ning Hsu**[‡*]    **James Glass**[†*]

[†] MIT CSAIL    [‡] Meta AI

alexhliu@mit.edu

## Abstract

In this paper, we introduce self-**di**stillatio**n** and **o**nline clustering for self-supervised **s**peech **r**epresentation learning (DinoSR) which combines masked language modeling, self-distillation, and online clustering. We show that these concepts complement each other and result in a strong representation learning model for speech. DinoSR first extracts contextualized embeddings from the input audio with a teacher network, then runs an online clustering system on the embeddings to yield a machine-discovered phone inventory, and finally uses the discretized tokens to guide a student network. We show that DinoSR surpasses previous state-of-the-art performance in several downstream tasks, and provide a detailed analysis of the model and the learned discrete units. Code available at https://github.com/Alexander-H-Liu/dinosr.

## 1 Introduction

Self-supervised speech representation learning techniques have been a game changer in recent years. Learning from unlabeled data has been shown to be effective for many downstream tasks such as speech recognition, translation, and language modeling [1, 2]. Among the flourishing self-supervised learning techniques, we are particularly interested in three methods: masked language modeling, self-distillation, and clustering.

Masked language modeling (MLM; [3]) predicts the masked part of a sentence based on the unmasked context and was first developed for training language models with bidirectional self-attention models [4]. The strong performance in various natural language processing tasks has enabled representation learning with MLM to quickly succeed in the field. Unsurprisingly, the MLM concept also applies to speech [5, 6] as it shares a similar structure to text in a more complex form.

Self-distillation representation learning has recently come into the spotlight with outstanding results for computer vision [7, 8] and speech tasks [9]. In contrast to the conventional supervised knowledge distillation method [10], self-supervised distillation does not require labeled data to train a teacher model to guide the student model. Instead, both models are trained with unlabeled data using paired relations augmented by data augmentation [7] or masking [9].

Clustering algorithms like K-means have been well-known unsupervised techniques long before deep learning methods arose. In the deep learning era, researchers have found clustering mechanisms beneficial to self-supervised models in a differentiable form known as vector quantization [11]. Driven by the nature of speech, which is a continuous signal containing a spoken form of discrete text, vector quantization is an ideal match for representation learning as many studies [12, 13, 5]

---

[*]Equal advising

37th Conference on Neural Information Processing Systems (NeurIPS 2023).

have discovered. Besides serving as an information bottleneck that filters out unnecessary content in high-dimensional spaces and improves performance, clustering also provides a glimpse of the characteristic of the latent embedding produced by the model by categorizing them [14].

In this paper, we introduce self-**di**stillatio**n** and **o**nline clustering for **s**elf-supervised **s**peech **r**epresentation learning (DinoSR) which leverages the positive aspects of the aforementioned methods. We show that these concepts complement each other and result in a strong representation learning model for speech. In brief, DinoSR first extracts contextualized embeddings from the input audio with a teacher network, then runs an online clustering system on the embeddings to yield a machine-discovered phone inventory, and finally uses the discretized tokens to guide the student network. Quantitatively, DinoSR surpasses the state-of-the-art in speech recognition with limited resources on LibriSpeech [15] and unsupervised acoustic unit discovery [16]. Moreover, DinoSR demonstrates strong interpretability by discretizing the high-dimensional embedding space into clusters closely aligned to human-defined phonetic units.

## 2   Related Work

Self-supervised speech representation learning with deep neural networks first emerged in the form of autoregressive models [11, 17–19] where the goal is to predict the future based on past observations. Subsequently, bidirectional models [5, 13, 20–22] relaxed the unidirectional limitation to achieve better results. A common learning paradigm for bidirectional models is MLM – masking part of the input and training the model to recover the missing information using unmasked targets. These targets can be derived from the audio signal using different strategies, such as surface features [5] or contrastive learning [13].

Following the MLM training scheme, HuBERT [20] proposed targeting discrete units generated by vanilla acoustic unit discovery systems. Such a system can be as simple as K-means clustering over MFCC features, or even random linear projections over spectrograms [23]. Interestingly, HuBERT found that the acoustic unit discovery system can be iteratively refined by running offline K-Means clustering on the output of a specific layer of the pre-trained model. However, several important hyper-parameters are required to obtain the best performance, such as the number of updates, the layer whose output is to be clustered, and the number of clusters for each iteration. While the proposed method is conceptually similar to HuBERT – MLM with discovered acoustic units, our method can be trained end-to-end with fewer heuristics by leveraging the self-distillation framework and online clustering.

Our method is also closely related to self-distillation methods for representation learning. These methods originated from image representation learning [7, 8], training a pair of identical models named student and teacher networks. The key to this framework is to provide different views of the same input by image augmentation to each model, and also to update them in different policies – gradient descent for the student model and exponential moving average for the teacher model. Following the self-distillation framework, Baevski et al. [9] generalized the method to speech processing by replacing image augmentation with the MLM masking strategy and found it effective. The key difference between this work and prior work is the online clustering mechanism that derives discrete targets instead of using continuous embeddings from the teacher model as targets. We also note that our method differs from studies in knowledge distillation from pre-trained speech representation models [24–27] which focus on inference efficiency and model compression.

## 3   Method

### 3.1   Self-distillation Paradigm

As illustrated in Figure 1, our method shares the same framework as recent self-supervised learning methods with self-distillation such as DINO [8]. The goal is to train a student network $\theta_{student}$ guided by a teacher network $\theta_{teacher}$ where both models share the same architecture, which, in our work, is a $K$-layer transformer encoder [4]. The teacher network in the self-distillation framework is simply a copy of the randomly initialized student network at the beginning of training.

To train the framework, we need to generate different *views* of the same input data for each model to avoid a trivial solution ($\theta_{student} = \theta_{teacher}$). While this is often done by data augmentation in computer

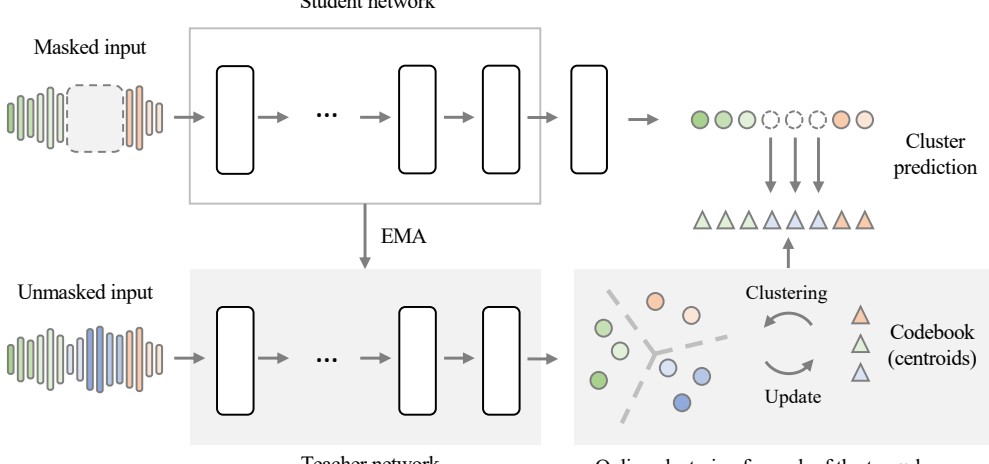

Figure 1: An overview of DinoSR: the teacher network is an exponential moving average of the student network and takes unmasked speech as input to extract target features. Online clustering is applied to multiple layers of the teacher, each with a separate codebook. The student network is trained to predict the corresponding clusters of masked input. Both teacher network and online clustering (shadowed regions) do not require gradients.

vision, we followed Baevski et al. [9] to use input masking as an alternative for speech. The input speech is partially masked for the student model to generate the masked representation $\mathbf{z}_t^K \in \mathbb{R}^D$ where $t = 1, ..., T$ is the sequence length. For the teacher model, the input is unmasked, and we denote the output representation $\tilde{\mathbf{z}}_t^K$.

Besides the different views of the same input, the parameter update policies of the two models are also different. While the student network is updated with gradient descent (with an objective function detailed later in §3.2), the teacher network parameter is updated via tracking the student network parameter with an exponential moving average (EMA):

$$\theta_{\text{teacher}} \longleftarrow \lambda\,\theta_{\text{teacher}} + (1 - \lambda)\,\theta_{\text{student}}, \tag{1}$$

where $\lambda$ is the decay rate of the teacher model in each training step.

## 3.2   Self-supervised Learning with DinoSR

**Acoustic Unit Discovery with Online Clustering.**   Under the self-distillation framework, our key contribution is to derive a good target from the teacher network to guide the student network. Prior work on self-supervised speech representation investigated acoustic unit discovery by either performing offline clustering of contextualized representations [20] or online clustering of non-contextualized representations [13]. DinoSR uses an online acoustic unit discovery system on top of the teacher network, providing contextualized discrete units. Unlike prior work using K-means clustering over MFCC features or pre-trained representations, our model's unit discovery system cannot be fixed since the teacher model evolves with the student model. As a solution, we propose performing online clustering at multiple layers of the teacher network.

For the $k$-th layer of the teacher model within the top $N$ layers (i.e., $k \in (K - N, K]$), we introduce a codebook (set of centroids) $\mathbf{E}^k = \{\mathbf{e}_1^k, ..., \mathbf{e}_V^k\}$ with $V$ codewords (centroids) $\mathbf{e}_i^k \in \mathbb{R}^D$. We update the codebook as follows: for each codebook entry $v$, we first create a set $\tilde{\mathbf{Z}}_v^k$ of the teacher output frames closest to the current representation of $v$ as per the codebook

$$\tilde{\mathbf{Z}}_v^k = \left\{ \tilde{\mathbf{z}}_t^k \;\middle|\; v = \operatorname*{argmin}_{i \in V} \left\| \tilde{\mathbf{z}}_t^k - \mathbf{e}_i^k \right\|_2 \right\}, \tag{2}$$

where the set index $v$ will be used as a pseudo label to train the student model. Each codeword is then updated using a weighted sum of the embeddings in this set using EMA:

$$
\begin{aligned}
\mathbf{s}_v^k &\longleftarrow \tau \, \mathbf{s}_v^k + (1 - \tau) \sum \tilde{\mathbf{Z}}_v^k, \\
n_v^k &\longleftarrow \tau \, n_v^k + (1 - \tau) \left| \tilde{\mathbf{Z}}_v^k \right|, \\
\mathbf{e}_v^k &\longleftarrow \frac{\mathbf{s}_v^k}{n_v^k}.
\end{aligned}
\tag{3}
$$

For each codeword $\mathbf{e}_v^k$, the first term $\mathbf{s}_v^k$ tracks the sum of all neighboring teacher representations (i.e., $\tilde{\mathbf{Z}}_v^k$ from Eq. 2), and the second term $n_v^k$ tracks the amount of the neighbors. With both terms approximated by EMA using the decay rate $\tau$, we have the codeword $\mathbf{e}_v^k$ which is the moving average of its neighbor set. In practice, we found performing online clustering on the subset of the frames where $t \in M$ is effective while reducing computation. For initialization, we set $\mathbf{s}_v^k$ to $\mathbf{e}_v^k$ and $n_v^k$ to 1. More details and discussions on online clustering are available in §A.2.

Since we define codewords by their neighboring representations, we can treat codewords as acoustic units discovered from the teacher model in an unsupervised manner and use them for training the student network. The clustering process creates discrete labels for frames based on their context in an end-to-end fashion. In §4.6, we show that these codewords possess similarities to human-defined acoustic units.

**Online Clustering v.s. Vector Quantization.** Van Den Oord et al. [11] first introduced vector quantization (VQ) to speech representation learning, encoding input audio signals into a sequence of discrete units. Later studies [28, 18, 14, 5] found that discretizing embedding spaces not only reduced the dimensionality of the model but also lead to performance improvements in downstream tasks. Another benefit of VQ to speech representation learning is better model interpretability. Previous work [12, 29] showed that the discretized representation could be viewed as model-discovered acoustic units which often aligned with human-defined units such as phonemes.

While there are similarities between VQ and the online clustering mechanism introduced here, they are also conceptually different. Prior works [18, 12, 13, 29] adopted VQ layer to serve as an efficacious discrete information bottleneck in the forward pass of the model; DinoSR leverages online clustering on gradient-free embedding space of the teacher model to mine acoustic units that can be treated as pseudo-labels. The most significant advantages of our method are 1) reducing computational costs; 2) bypassing estimations that are required by the non-differentiable nature of VQ, e.g., approximating the gradient with straight-through gradient estimator [30]; 3) mitigating problems in practice such as code collapse as shown in §4.6.

**Self-supervised Learning via Cluster Prediction** For each output frame of the student model $\mathbf{z}_t^K$, the training objective is to predict the codeword index $v$ of the corresponding frame from the teacher model (i.e., $\tilde{\mathbf{z}}_t^k \in \tilde{\mathbf{Z}}_v^k$) across all targeted layers,

$$
\sum_{t \in M} \sum_{k \in (K-N, K]} \log p_{\phi_k}(v | \mathbf{z}_t^K),
\tag{4}
$$

where $M$ denotes the set of all masked timesteps and $\phi_k$ is the prediction head composed of a linear projection $\mathbb{R}^{D \times V}$ followed by a softmax activation for each target layer $k$. Note that the prediction head is at the last layer $K$ of the student model regardless of the target layer $k$. In §A.3, we summarize the pre-training of DinoSR with pseudo-code to provide a complete view of our method.

## 4 Experiments

### 4.1 Pre-training

Following Hsu et al. [20] and Baevski et al. [9], we use 960 hours of speech from the LibriSpeech [15] corpus to pre-train our model. We focus on the BASE sized transformer [4] with $K = 12$ layers and embedding dimension $D = 768$ due to resource constraints, with the batch size of 63 minutes of audio in total across 16 GPUs. The 16 kHz input waveform is first downsampled to 50Hz with a convolutional feature encoder [13]. For the student model, we randomly masked $M = 80\%$ of the 50Hz input features before feeding them into the transformer, with each masked span no shorter than

Table 1: Acoustic unit discovery results on ZeroSpeech 2021 challenge [16] in ABX error rate.

| Method | Target layer | same-speaker | | cross-speaker | | Average |
|---|---|---|---|---|---|---|
| | | clean | other | clean | other | |
| **Best challenge participants**[1] | | | | | | |
| Nguyen et al. [32] | - | 3.26 | 3.81 | 4.00 | 5.91 | 4.25 |
| Chorowski et al. [33] | - | **2.95** | 3.54 | 4.50 | 7.05 | 4.51 |
| **Self-supervised speech representation models**[2] | | | | | | |
| wav2vec 2.0 [13] | 6 | 4.15 | 5.22 | 4.82 | 7.38 | 5.39 |
| HuBERT [20] | 11 | 3.07 | 3.90 | 3.71 | 6.19 | 4.22 |
| data2vec [9] | 4 | 4.03 | 5.09 | 4.72 | 6.97 | 5.20 |
| ContentVec [34] | 12 | 2.98 | 3.70 | 3.44 | 5.17 | 3.82 |
| DinoSR | 5 | 3.08 | **3.43** | **3.42** | **4.42** | **3.59** |

[1] Results from `https://zerospeech.com/tasks/task_1/results/`
[2] Evaluating official model released by the authors.

10 frames. For the teacher model, the input feature is not masked, and online clustering is performed at the top $N = 8$ layers (i.e., $k \in [5, 12]$), each with a codebook with $V = 256$ codewords. The codebook decay rate $\tau$ is fixed at 0.9.

The student model is trained for 400k steps with the Adam optimizer [31] with a learning rate ramped up linearly to 0.0005 within the first 12k steps, held for the following 188k steps, and exponentially decayed to 0.00005 for the final 200k steps. The teacher model decay rate $\lambda$ increases linearly from 0.999 to 0.9999 within the first 30k updates, held for the next 200k steps, and increased to 1.0 for the remaining steps. Pre-training the model takes about 180 hours on 16 Nvidia V100 GPUs. After pre-training, the student model is evaluated on different downstream tasks.

### 4.2 Acoustic Unit Discovery

To examine the effectiveness of the online clustering mechanism used in DinoSR, we consider the acoustic unit discovery benchmark introduced in the Zero Resource Speech Challenge 2021 [16]. In this task, the speech representation extracted from a frozen pre-trained model is used for unit discovery. The task is an ABX discrimination test: given a pair of spoken triphones (A and B, e.g., 'aba' and 'apa'), the model must decide which triphone a new input (X, e.g., 'apa') corresponds to. The new triphone can be spoken by the same speaker as A and B in the same-speaker setup, or by a different speaker in a more challenging cross-speaker setup. The evaluation metric is the decision error rate on the dev set.

To measure the similarity between two sequences of a speech representation, the task introduced a pseudo-distance defined as the average framewise distance over the dynamic time warping path. A common choice of framewise distance is the cosine distance between two embedding vectors. Different from cosine similarity, we define the framewise distance as the JS-divergence between framewise probability over the codebook as defined in Eq. 4 to take advantage of the learned discrete units.

Results are shown in Table 1 with three important observations. First, it can be shown that previous self-supervised methods do not surpass methods specialized for acoustic unit discovery [33, 32]. DinoSR, however, outperforms all other methods by a margin except in the easiest same-speaker clean-speech setup. Second, DinoSR performs better than HuBERT, which also leverages representation clustering for training. Finally, in this task, the continuous self-distillation method data2vec lags both DinoSR and HuBERT. With these observations, we conclude that the codebook design in DinoSR is effective for audio clustering, leading to its superior performance in acoustic unit discovery.

### 4.3 Fine-tuning DinoSR for Speech Recognition

Following the protocol proposed by Baevski et al. [13] and adopted by prior work [20, 22, 9], we fine-tune the student model using CTC [35] using labeled speech data under four different setups, using 10 minutes / 1 hour / 10 hours from LibriLight [36] or 100 hours from LibriSpeech [15]. After fine-tuning, we measure the word error rate (WER) on LibriSpeech by decoding test sets using the

Table 2: Word Error Rate (WER) on LibriSpeech standard dev/test sets. All models are BASE size (12-layer) transformer encoders pre-trained on the full LibriSpeech dataset (960 hours) and decoded with 4-gram language model. The best result in each setup is **bolded** and the second best is underlined.

| Model | Pre-training steps | Batch size (minutes) | dev | | test | |
|---|---|---|---|---|---|---|
| | | | clean | other | clean | other |
| **10 minutes labeled data** | | | | | | |
| wav2vec 2.0 [13] | 400k | 96 | 8.9 | 15.7 | 9.1 | 15.6 |
| HuBERT [20] | 250k + 400k | 47 | 9.1 | 15.0 | 9.7 | 15.3 |
| data2vec [9] | 400k | 63 | 7.3 | 11.6 | 7.9 | 12.3 |
| DinoSR | 400k | 63 | **6.6** | **10.8** | **7.3** | **11.8** |
| **1 hr labeled data** | | | | | | |
| wav2vec 2.0 [13] | 400k | 96 | 5.0 | 10.8 | 5.5 | 11.3 |
| HuBERT [20] | 250k + 400k | 47 | 5.6 | 10.9 | 6.1 | 11.3 |
| WavLM [22] | 250k + 400k | 187 | - | - | 5.7 | 10.8 |
| data2vec [9] | 400k | 63 | **4.0** | 8.5 | **4.6** | 9.1 |
| DinoSR | 400k | 63 | 4.1 | **8.1** | **4.6** | **8.7** |
| **10 hr labeled data** | | | | | | |
| wav2vec 2.0 [13] | 400k | 96 | 3.8 | 9.1 | 4.3 | 9.5 |
| HuBERT [20] | 250k + 400k | 47 | 3.9 | 9.0 | 4.3 | 9.4 |
| WavLM [22] | 250k + 400k | 187 | - | - | 4.3 | 9.2 |
| data2vec [9] | 400k | 63 | 3.3 | 7.5 | 3.9 | 8.1 |
| DinoSR | 400k | 63 | **3.1** | **7.0** | **3.6** | **7.6** |
| **100 hr labeled data** | | | | | | |
| wav2vec 2.0 [13] | 400k | 96 | 2.7 | 7.9 | 3.4 | 8.0 |
| HuBERT [20] | 250k + 400k | 47 | 2.7 | 7.8 | 3.4 | 8.1 |
| WavLM [22] | 250k + 400k | 187 | - | - | 3.4 | 7.7 |
| data2vec [9] | 400k | 63 | **2.2** | **6.4** | **2.8** | 6.8 |
| DinoSR | 400k | 63 | 2.3 | **6.4** | 2.9 | **6.7** |

official 4-gram language model. The decoding hyper-parameter is searched with Ax [2] following the prior works.

We compare DinoSR to four recent works that all adopted MLM with the BASE sized transformer, and followed the same fine-tuning regime: 1) wav2vec 2.0 [13], a method relying on contrastive learning with VQ over local target representations; 2) HuBERT [20], an iterative method with offline clustering over global target representations; 3) WavLM [22], an iterative method guided by 1st iteration HuBERT and an auxiliary denoising task; and 4) data2vec [9], a self-distillation method with regression loss over contextualized target representations. In Table 2, we summarize the results and compare them to prior work using the same setup. We also list the total pre-training steps and batch size used for each method to indicate the computation needed.

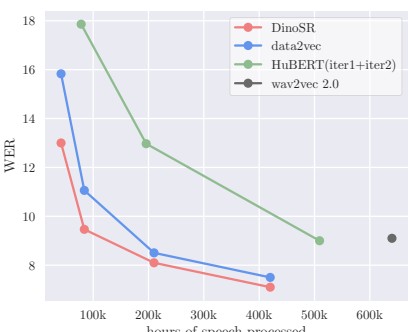

Figure 2: The trade-off between performance (WER on LibriSpeech dev-other) and data efficiency (hours of speech the model processed in total during pre-training) for different methods.

Compared to other methods that rely on discrete units, our method is significantly stronger while reducing the batch size (vs. contrastive method wav2vec 2.0) and the training steps (vs. iterative offline clustering methods HuBERT and WavLM). This demonstrates the advantage of learning discrete units with online clustering instead of contrastive learning or offline clustering. An improvement over data2vec, the previous state-of-the-art method, is observed in most setups. This result shows that using discrete units as a learning target benefits speech representation learning. Despite being on par or slightly worse in a few setups, this

---

[2] https://github.com/facebook/Ax

Table 3: Results on Speech Processing Universal PERformance Benchmark [37] (SUPERB). The tasks include phoneme recognition (PR), automatic speech recognition (ASR), keyword spotting (KS), intent classification (IC), slot filling (SF), and speech translation (ST). Metrics include accuracy (Acc%), phoneme error rate (PER%), word error rate (WER%), F1 score (F1%), concept error rate (CER%), and bilingual evaluation understudy score (BLEU). The best result in each task is **bolded** and the second best is underlined.

| Model[1] | Content | | | Semantic | | | |
|---|---|---|---|---|---|---|---|
| | PR | ASR | KS | IC | SF | | ST |
| | PER↓ | WER↓ | Acc↑ | Acc↑ | F1↑ | CER↓ | BLEU↑ |
| wav2vec 2.0 [13] | 5.74 | 6.43 | 96.23 | 92.35 | 88.30 | 24.77 | 14.81 |
| CCC-wav2vec 2.0 [38] | 5.95 | 6.30 | 96.72 | 96.47 | 88.08 | 24.34 | 16.20 |
| HuBERT[2] [20] | 5.41 | 6.42 | 96.30 | 98.34 | 88.53 | 25.20 | 15.53 |
| WavLM[2,3] [22] | 4.84 | 6.31 | **96.79** | **98.63** | **89.38** | **22.86** | **20.74** |
| data2vec [9] | 4.69 | 4.94 | 96.56 | 97.63 | 88.59 | 25.27 | 17.42 |
| DinoSR | **3.21** | **4.71** | 96.69 | 98.02 | 88.83 | 23.57 | 17.68 |

[1] Best models on leaderboard. For a complete comparison, please visit `https://superbbenchmark.org/leaderboard`.
[2] Training requires iterative offline clustering, see §4.3 and Table 2.
[3] State-of-the-art with large batch size and hyper-parameter sweep in all downstream tasks, see §4.4 and Table 2.

benchmark has been thoroughly studied; thus, progress is not easily attained. Moreover, we show that DinoSR consistently outperforms data2vec in other benchmarks later in this section.

Beyond recognition performance, we examined the data efficiency of each method, as shown in Figure 4.3. We introduce the metric *hours of speech processed* that reflects the amount of speech one model needs to "hear" during pre-training. The metric is defined as the number of updates required to train the model × batch size in hours, using attributes available in Table 2. By comparing DinoSR against prior work, we see the advantage of being more data efficient, requiring less training yet performing better.

## 4.4 Downstream Evaluation

We further evaluate the effectiveness of DinoSR representations using the Speech Processing Universal PERformance Benchmark (SUPERB) [37, 39]. SUPERB is a benchmark consisting of ten speech-processing tasks spanning content, semantics, speaker, and paralinguistics tasks. To better understand the capabilities of modeling content and semantics, we report the results of our model on phoneme recognition (PR), automatic speech recognition (ASR), keyword spotting (KS), intent classification (IC), slot filling (SF), and speech translation (ST).

In SUPERB, each pre-trained SSL model is frozen and serves as a feature extractor. In each task, a set of learnable weights are used for weighted-summing all layers' features. Then, the weighted-summed features are fed into a lightweight prediction head to generate outputs. Thus, only the learnable weights and the prediction head are fine-tuned with labeled data.

The SUPERB results are shown in Table 3. In content tasks, the DinoSR surpasses prior art on PR and ASR, showing its capability of capturing better phonetic information. For semantic tasks like IC and SF, DinoSR has similar performance as WavLM [22] and HuBERT.

Though DinoSR falls slightly behind the state-of-the-art model WavLM on SUPERB, it is worth pointing out that WavLM is a second iteration model based on HuBERT with a large batch size, requiring significantly more computational resources for pre-training. Moreover, WavLM has done a hyper-parameter search for each task in SUPERB (see Appendix A in Chen et al. [22]) whereas DinoSR is tested with no more than five runs in each downstream task due to resource limitations.

## 4.5 Impact of Codebook Hyper-parameters

To study the impact of several hyper-parameters used by DinoSR, we vary different options, including the codebook size $V$ (default 8), the top $N$ layers to apply online clustering (default 8), and the

codebook decay rate of $\tau$ (default 0.9). To reduce computation, we use the 10-hour subset to fine-tune the teacher network after 200k steps of pre-training. WERs are reported by decoding the dev-other subset with a fixed language model weight of 2, and word insertion penalty of $-1$, following Baevski et al. [13]. Results are presented in Figure 3 and Table 4.

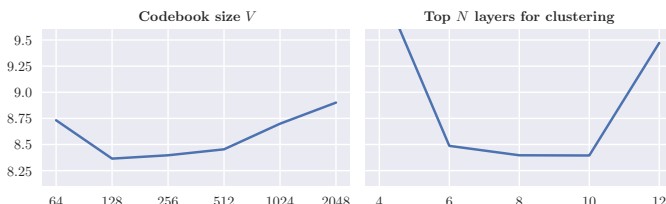

Figure 3: Varying codebook size $V$ and the number of codebooks $N$.

Table 4: Varying code-book decay $\tau$.

| $\tau$ | WER |
|---|---|
| 0.5 | 8.57 |
| 0.6 | 8.30 |
| 0.7 | 8.54 |
| 0.8 | 8.88 |
| 0.9 | 8.40 |
| 0.99 | 8.73 |
| 0.999 | 9.43 |
| $0.9 \longrightarrow 0.99$ | 8.71 |
| $0.9 \longrightarrow 0.999$ | 8.60 |

Surprisingly, varying the codebook size $V$ from 64 to 2048 only changed the resulting WER by a small margin. Compared to codebook size $V$, the choice of the top $N$ layers to cluster has a larger impact on the results, with the best choices ranging from 6 to 10. For the codebook decay rate $\tau$, we found values between 0.5 to 0.99 worked well in general. Since the teacher network decay $\lambda$ anneals throughout the training, we also tested and found annealing the codebook decay $\tau$ to 0.99 or 0.999 is unnecessary. We suspect the stability originates from the slow-changing property of the teacher network updated via EMA.

## 4.6 Analysis

In this section, we took a closer look at the properties of the discrete units. We focused on the fifth layer of DinoSR and leave more analysis and comparisons against prior works in the appendix §A.4.

**Cluster quality.** To measure the quality of the discrete units learned by DinoSR, we adopt the three metrics proposed in HuBERT[20] as well as codebook perplexity [40]:

- *Cluster purity* (Cls Pur.) measures the purity of the set of associated codewords of each phone.
- *Phone purity* (Phn Pur.) measures the purity of the set of associated phones of each codeword.
- *Phone-normalized mutual information* (PNMI) measures the uncertainty reduction for the underlying phone when observing the codeword of a frame.
- *Codebook perplexity* (Code Ppl.) $2^{-\sum_V p(v)\log_2 p(v)}$ measures the diversity of codewords being used by the model with $p(v)$ being the frequency distribution over the dataset. For example, code ppl.$=$ codebook size indicates all codewords are being used equally.

To compute these metrics, forced alignment is used to acquire the ground truth phone of each feature frame on LibriSpeech dev-clean and dev-other sets. The maximum cluster size for all methods is fixed to 256 for a fair comparison except VQ-APC [12]. Note that for online clustering methods, the number of active clusters might be lower due to the defect of vector quantization, and we report the number of active clusters. VQ-APC suffers from code collapse with vanilla VQ which leads to lower code usage and code ppl., so we use the model with a larger 512 codewords instead. Co-training APC [29] can be viewed as an improved version of VQ-APC which solved the problem by penalizing low codebook perplexity during training. Wav2vec 2.0 [13] is not applicable to this test since it used multiple codebooks that partitioned feature dimensions into 16 groups. Results are listed in Table 5.

The MFCC clusters, which are used to train the first iteration HuBERT, provided a baseline for purity and PNMI. The first and second iterations of HuBERT, which served as the teacher in HuBERT's iterative pre-training procedure, show a significant improvement over MFCCs. The results show performing K-means clustering on DinoSR, which does not require an iterative process, produces slightly better quality clusters. DinoSR makes better use of codewords compared to prior VQ works, having 217 active clusters out of 256 despite running online clustering. Better codebook usage results in a notable improvement in cluster quality since each cluster can be finer-grained. DinoSR achieved a comparable phone purity and PNMI compared to offline methods while being more efficient. Interestingly, the codebook's cluster purity surpasses offline clustering methods, which further supports the effectiveness of the proposed method.

Table 5: Discrete unit quality on LibriSpeech dev set measured by Codebook Perplexity (Code Ppl.), Cluster purity (Cls Pur.), Phone purity (Phn Pur.), and Phone-normalized mutual information (PNMI). Results are compared to HuBERT [20], VQ-APC [12], and co-training APC [29] using code and models released by the authors.

| Method | Active cluster | Code Ppl. | Cls Pur. | Phn Pur. | PNMI |
|---|---|---|---|---|---|
| K-means (offline clustering) | | | | | |
| MFCC | 256 | 228.2 | 0.06 | 0.30 | 0.28 |
| HuBERT-iter1 L6 | 256 | 231.8 | 0.15 | 0.60 | 0.60 |
| HuBERT-iter2 L9 | 256 | 228.6 | 0.15 | 0.61 | 0.61 |
| DinoSR L5 | 256 | 242.4 | **0.17** | **0.63** | **0.62** |
| Codebook (online clustering) | | | | | |
| VQ-APC | 98 | 72.1 | 0.08 | 0.24 | 0.19 |
| Co-training APC | 164 | 135.0 | 0.09 | 0.31 | 0.29 |
| DinoSR L5 | 217 | 179.2 | **0.19** | **0.58** | **0.57** |

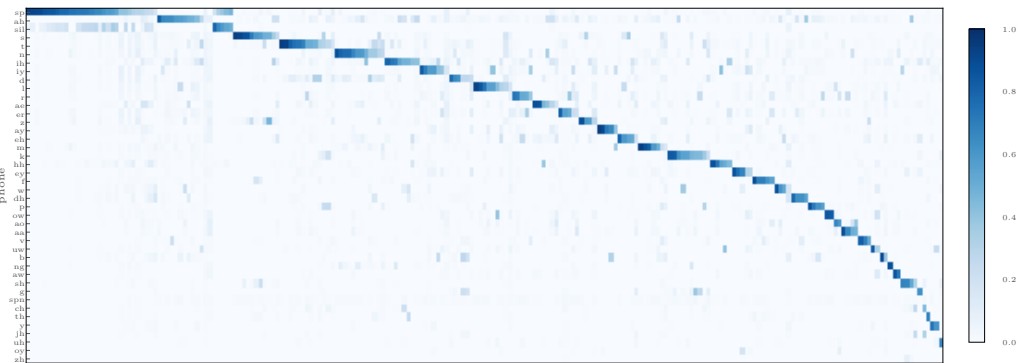

Figure 4: The conditional probability $P(\text{phone}|\text{code})$ on LibriSpeech dev set visualized. The y-axis is the phone set sorted by the number of occurrences, the x-axis is the 217 active codewords sorted by the most correlated phone. A larger figure for clarity is provided in §A.4.

**Mapping phones to codewords.** To demonstrate the quality of the learned codebook, we visualize the conditional probability $P(\text{phone}|\text{code})$ accumulated over the LibriSpeech dev sets in Figure 4.

We highlight two interesting findings: 1) Each codeword is typically concentrated on one phone, reflecting the high phone purity obtained in the quality test. In the case where two phones shared high usage of the same codeword, we observed the sharing phones are acoustically similar such as /sp/ (short pause) and /sil/ (silence) in the upper left corner. 2) The overall usage of codewords captures the long-tail nature of phone distribution. The more frequent phones (upper part in figure) occupied significantly more codewords. The top 10 most frequent phones (/sp/ to /L/) held over 50% of the active codewords. This phenomenon, again, supports our claim that the proposed online clustering method is a good acoustic unit discovery system. As a reference, using the mapping for classification (by assigning each codeword to the dominant phone and treating all other phones assigned to the codeword as error) in the figure results in a frame-wised phone error rate of 58.2%. Additional visualization of the discrete embedding can be found in Section A.5.

## 5 Conclusion

In this paper, we introduced DinoSR – a new self-supervised method motivated by the continuous-to-discrete nature of speech understanding, leveraging recent advances in representation learning. The key innovation of DinoSR is to introduce a gradient-free online clustering method that leads to meaningful acoustic units. Our main contributions include advancing the state-of-the-art in different benchmarks with end-to-end training and providing a closer look at embeddings from speech transformers via the discrete unit. Future work includes structural learning with the codebook, scaling to larger models, and extending the model to different modalities.

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

# A  Appendix

## A.1  Broader Impact and Limitations

This work, alone with the related prior works in self-supervised speech representation learning discussed, focused on English speech, thereby implying the potential risks associated with neglecting other languages, especially the low-resource languages. Nevertheless, we provide a preliminary result on other languages in §A.6 to mitigate the problem as much as possible. In addition, the design of DinoSR focused on learning acoustic units to models the phonetic contents in speech. Consequently, other contents (e.g., speaker/paralinguistic information) might be neglected by the model.

## A.2  Discussion on online clustering

**Codebook initialization.**  The initialization of codebook $\mathbf{E}^k \in \mathbb{R}^{V \times D}$ can be critical in vector quantization methods [11]. We tried different initializations including

- $\mathbf{E}^k \sim \mathcal{N}(0, \mu)$ where $\mu \in \{1, \frac{1}{\sqrt{D}}\}$,

- $\mathbf{E}^k \sim \mathcal{N}(0, 1)$ followed by L2 normalization $\frac{\mathbf{e}_v^k}{\|\mathbf{e}_v^k\|_2}$.

In practice, we found different initialization leads to different codebook perplexity at the beginning of training. Nevertheless, the methods all lead to similar codebook perplexity at the end of training and downstream performance. This also demonstrated the stability of gradient-free online VQ in oppose to standard VQ.

**Input for quantization.**  We found normalizing the teacher model representation $\tilde{\mathbf{z}}_t^k$ is necessary for stable clustering. We briefly summarized the results using different normalization methods:

- Instance normalization (IN; default): this can be interpreted as a parameter-free utterance-wise normalization for each channel, we found it stable.
- Batch normalization (BN): this can be viewed as a dataset-wise version of IN which yields a similar result but introduces additional parameters tracking the running stats.
- L2 normalization: a frame-wise normalization along the feature dimension, results in a more unstable codebook perplexity and code collapse occasionally.

In addition, we also tried combining targets from different layers (i.e, $\sum_k \tilde{\mathbf{z}}_t^k$ with a single codebook across all layers) before clustering following Baevski et al. [9]. This model performed significantly worse in the downstream fine-tuning task.

**Dealing with inactive codewords.**  Note that the codebook update policy

$$
\begin{aligned}
\mathbf{s}_v^k &\longleftarrow \tau\,\mathbf{s}_v^k + (1-\tau)\sum \tilde{\mathbf{Z}}_v^k, \\
n_v^k &\longleftarrow \tau\,n_v^k + (1-\tau)\left|\tilde{\mathbf{Z}}_v^k\right|, \\
\mathbf{e}_v^k &\longleftarrow \frac{\mathbf{s}_v^k}{n_v^k},
\end{aligned}
\tag{3}
$$

updates each codeword $\mathbf{e}_v^k$ regardless of whether the codeword activate (i.e., having at least one neighbor $\left|\tilde{\mathbf{Z}}_v^k\right| \geq 1$) or not. In the extreme case where a codeword remains inactive for a long period, we have $\mathbf{s}_v^k \to \vec{0}$ and $n_v^k \to 0$ which results in numerical instability or code collapse. In practice, we found freezing the inactivate codewords with

$$
\tau_v^k = \begin{cases} \tau, & \text{if } \left|\tilde{\mathbf{Z}}_v^k\right| \geq 1 \\ 1, & \text{otherwise} \end{cases},
\tag{5}
$$

leads to a slightly better codebook perplexity but the improvement diminishes as the batch size increases.

**Simplifying codeword update policy.** A simplified version of online clustering described in Eq.3 is to only track the averaged embedding without the size

$$\mathbf{e}_v^k \longleftarrow \tau \, \mathbf{e}_v^k + (1 - \tau) \frac{\sum \tilde{\mathbf{Z}}_v^k}{\left| \tilde{\mathbf{Z}}_v^k \right|}.$$

(6)

The simplified version enforces equal momentum for each step regardless of the size of the neighbor set $\left| \tilde{\mathbf{Z}}_v \right|$. In practice, we found using Eq. 3 more stable and results in slightly better performance with a negligible cost.

## A.3 Pseudo-code for DinoSR training

---

**Algorithm 1** PyTorch pseudocode for DinoSR

---

```
# teacher, student: student and teacher networks
# phi[k]: DxV cluster prediction matrix for k-th layer codebook
# codebook[k]: VxD codebook matrix for k-th layer
# code_sum[k]: VxD unnormalized codebook matrix for k-th layer
# code_cnt[k]: Vx1 codeword counter for k-th layer
# lbd, tau: decay rates of teacher network, codebook
teacher.weight = student.weight

for x in dataset: # mini audio batch BxT

    # Eq.1: teacher EMA
    teacher.weight = lbd * teacher.weight + \
                     (1-lbd) * student.weight

    z = student(mask(x))     # BxTxD, last layer only
    z = z[masked_position]   # MxD
    with torch.no_grad():    # gradient-free syntax
        z_tilde = teacher(x) # KxBxTxD, all K layers
        z_tilde = z_tilde[:,masked_position] # KxMxD

    loss = 0
    for k in range(K-N,K):

        with torch.no_grad():
            # Eq.2: online clustering
            d = -framewiseL2(z_tilde[k],codebook[k])
            target_cls = hardmax(d, dim=-1) # MxV

            # Eq.3: codebook learning
            code_sum[k] = tau * code_sum[k] + \
                    (1-tau) * matmul(target_cls.T,z_tilde)
            code_cnt[k] = tau * code_cnt[k] + \
                    (1-tau) * target_cls.sum(dim=0)
            codebook[k] = code_sum[k] / code_cnt[k]

        # Eq.4: cluster prediction
        p_v = phi[k](z) # MxV
        loss += cross_entropy(p_v, target_cls)

    loss.backward()
    student.step()
```

---

## A.4 Additional results and analysis

**Visualizing phone-code correlation.** To further demonstrate the difference between DinoSR and prior works with online codebook learning, we visualized the conditional probability $P(\text{phone}|\text{code})$ computed using Co-training APC [3][29] and Vector-Quantized Autoregressive Predictive Coding (VQ-APC[4];[12]) following the exact same setup used in Figure 4. Clearly, DinoSR is able to capture the long-tail distribution better, and the codewords tend to be more concentrated to the most correlated phone when compared against the prior works.

---

[3]Model checkpoint from `https://github.com/30stomercury/autoregressive-co-training`
[4]Code from `https://github.com/iamyuanchung/VQ-APC`

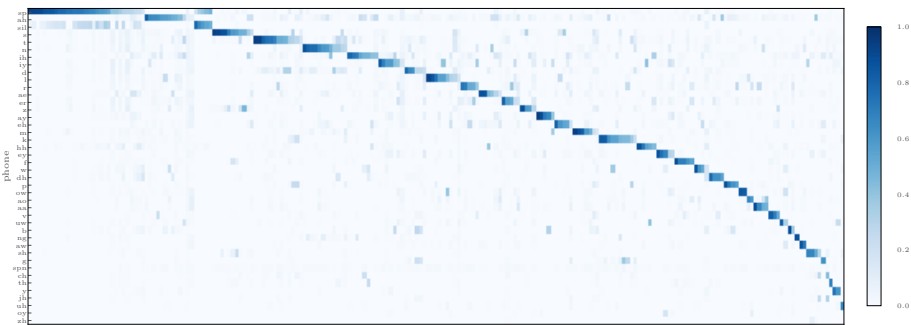

Figure 5: $P(\text{phone}|\text{code})$ from DinoSR with 217 codewords activated out of 256.

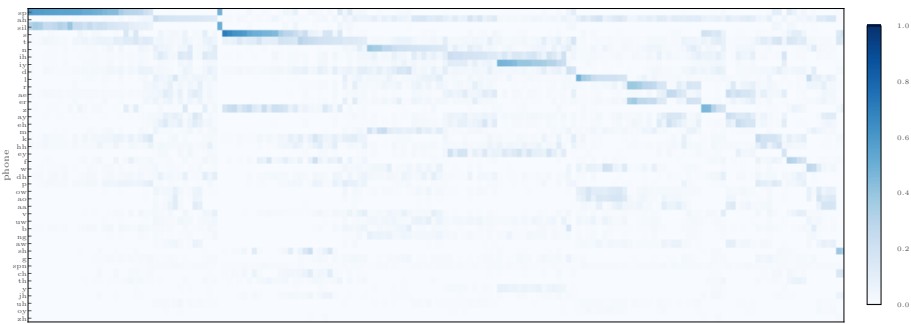

Figure 6: $P(\text{phone}|\text{code})$ from Co-training APC [29] with 164 codewords activated out of 256.

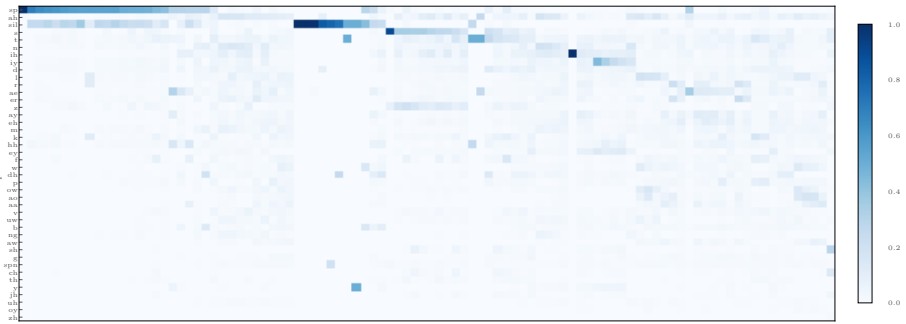

Figure 7: $P(\text{phone}|\text{code})$ from VQ-APC [12] with 98 codewords activated out of 512.

Figure 8 and figure 9 provided another view of the codeword distribution over the phone set. Each codeword is assigned to a single phone based on the most correlated phone (i.e., $\text{argmax}_{\text{phone}} P(\text{phone}|\text{code})$). We derive the learned phone distribution by accumulating the occurrence of all codewords and compare to the ground truth. Results show the distribution of codewords from DinoSR is very close to the ground truth, while other methods failed to capture the underlying distribution by over-assigning codewords to the more frequent phones and dropping the less frequent phones.

**Finding the best codebook.** By examining phone-normalized mutual information, code perplexity, and ABX score in acoustic unit discovery in each layer, we can see that the 5th layer of the teacher model consistently performed the best. However, this is only considering phones as the ideal discrete unit. A more throughout study on the content of each layer is left as future work.

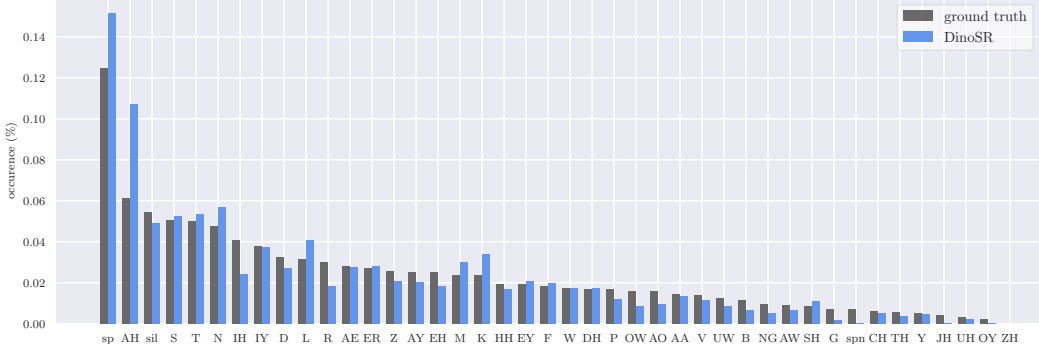

Figure 8: Histogram of phones and codewords.

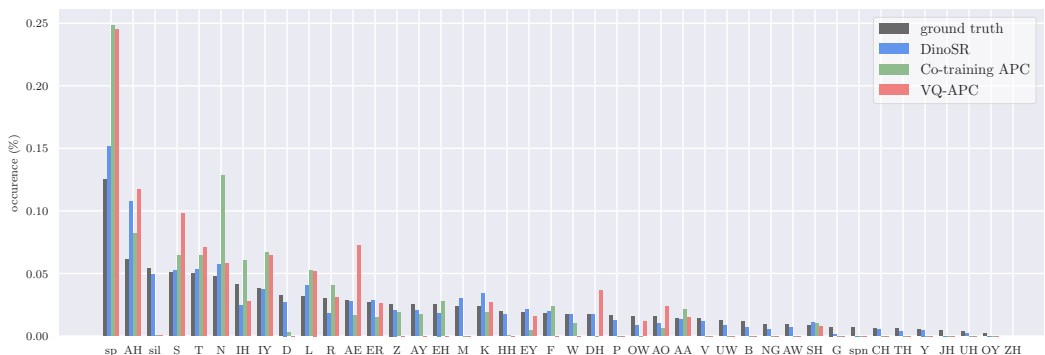

Figure 9: Histogram of phones and codewords.

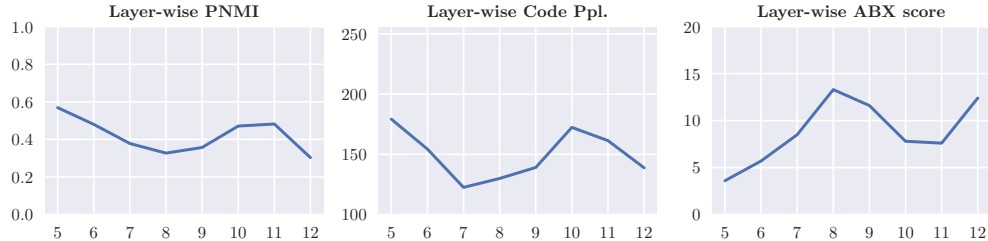

Figure 10: Layer-wise phone-normalized mutual information, code perplexity, and ABX score in acoustic unit discovery.

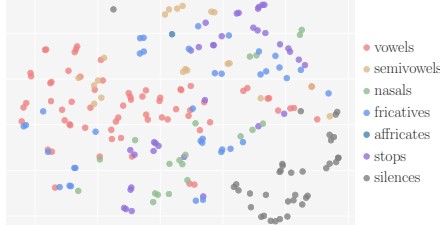

Figure 11: Visualizing codebook using t-SNE [41]. Each codeword is categorized into an articulation manner class by the most correlated phone.

## A.5 t-SNE visualization of codewords

Besides quantitative evaluations, we provide a qualitative result in Figure 11 by using t-SNE [41] to visualize the codebook in 2-dimensional space. By labeling each codeword using articulation manner classes in English, we revealed the fact that some of the acoustic attributes are embedded in

the high-dimensional space. For example, both vowels and slilences demonstrated a high degree of concentration.

## A.6   Preliminary Result on Multi-lingual Speech

We conducted a preliminary experiment under limited computing budget to showcase that our method can be generalized to other languages. We followed the setting in multi-lingual speech representation learning [42, 43] to pre-train DinoSR on 10 different languages (Bengali, Cantonese, Georgian, Haitian, Kurmanji, Pashto, Tamil, Turkish, Tokpisin, Vietnamese) on the BABEL dataset [44] and fine-tune on 4 different unseen languages (Assamese, Tagalog, Swahili, Lao). In this setup we trained our model for 200k steps on 4 GPUs with a total batch size of 16 minutes. We report Character Error Rate (CER) on the fine-tuning languages in Table 6.

Table 6: Character Error Rate (CER) on BABEL dataset.

| Model | Pre-training steps | Batch size (minutes) | Fine-tuning Language | | | |
| --- | --- | --- | --- | --- | --- | --- |
| | | | Assamese | Tagalog | Swahili | Lao |
| Seq-to-seq ASR without pre-training [42] | - | - | 41.3 | 37.9 | 29.1 | 38.7 |
| XLSR-10 [43] | 250k | 96 | 29.4 | 21.9 | 16.6 | 23.3 |
| DinoSR | 200k | 16 | 27.2 | 19.4 | 14.6 | 22.8 |

Inspired by the International Phonetic Alphabet [45] which defined a universal set of acoustic units across different languages, we take a look into how DinoSR acoustic units derived from the 10 pre-training languages are distributed. Interestingly, we found the acoustic units are more likely to be shared across different languages as shown in Table 7.

Table 7: Codeword distribution over different languages from BABEL dataset. Each cell corresponded to the proportion of codewords activated in $n$ languages, e.g., 86.5% of the codewords emerged in all 10 languages for the 5th layer.

| Layer | $n$ languages | | | | | | | | | |
| --- | --- | --- | --- | --- | --- | --- | --- | --- | --- | --- |
| | 1 | 2 | 3 | 4 | 5 | 6 | 7 | 8 | 9 | 10 |
| 5th | 0.0% | 0.0% | 4.5% | 1.7% | 1.1% | 0.6% | 1.1% | 3.4% | 1.1% | 86.5% |
| 6th | 0.0% | 0.0% | 2.9% | 1.7% | 1.7% | 1.7% | 1.7% | 1.7% | 2.9% | 85.5% |
| 7th | 0.7% | 0.7% | 4.0% | 1.3% | 1.3% | 0.7% | 3.4% | 4.0% | 7.4% | 76.5% |
| 8th | 0.0% | 0.0% | 1.9% | 2.6% | 1.3% | 0.6% | 1.3% | 2.6% | 1.9% | 87.8% |
| 9th | 0.0% | 0.0% | 2.4% | 1.8% | 1.8% | 1.2% | 0.6% | 1.2% | 3.6% | 87.4% |
| 10th | 0.0% | 0.0% | 1.8% | 1.2% | 1.2% | 0.0% | 0.6% | 1.2% | 1.8% | 92.1% |
| 11th | 0.0% | 0.0% | 1.9% | 0.6% | 0.0% | 0.6% | 0.0% | 2.5% | 1.9% | 92.5% |
| 12th | 0.0% | 0.0% | 0.0% | 0.0% | 0.0% | 0.0% | 0.6% | 1.2% | 0.6% | 97.6% |

In conclusion, preliminary results on BABEL show that DinoSR can be applied to languages other than English. Moreover, learning acoustic units from different languages is possible even with a shared vocabulary.

