# OpenReview forum: "DinoSR: Self-Distillation and Online Clustering for Self-supervised Speech Representation Learning"
_NeurIPS.cc/2023/Conference — NeurIPS 2023 poster_

### Official Review · Reviewer_RFky · 2023-06-30

**Soundness:** 3 good
**Presentation:** 4 excellent
**Contribution:** 3 good
**Rating:** 7
**Confidence:** 4

**Summary:**

The authors propose a self-supervised speech representation that combines masked language modeling, online clustering, and self-distillation. They apply these techniques using jointly-trained transformer-based teacher and student models, where the student has to guess the cluster assignment of masked input. Evaluation of the student model demonstrates a new state-of-the-art speech representation on key metrics such as phoneme error rate and word error rate. Evaluation of the learned clusters indicates some visual agreement with the phoneme set of the CMU pronunciation dictionary.

**Strengths:**

Having not worked with online clustering or self-distillation myself, I found the explanations provided in sections 1 to 3 to be an articulate and helpful introduction to the topics—as well as how they interact.

Figure 1 is a well-done visual aid for the proposed method.

Thorough evaluation on multiple relevant downstream tasks demonstrates clear improvements over state-of-the-art self-supervised speech representations. Figure 2 is an especially clear demonstration of the efficacy of the proposed representation.

Overall a well-organized and compelling argument for a new state-of-the-art speech representation.


**Weaknesses:**

Section 4.6 is well done, although the “mapping phones to codewords” section could be made more compelling by utilizing the mapping to perform phoneme classification and producing a phoneme error rate.

The bolding in table 5 confuses me, both because there are multiple bold items per column, and some columns have no bold items without a clear reason.


**Questions:**

Previously, open-source releases for this research task (e.g., wav2vec 2.0) have contributed significantly to downstream research efforts. Do you intend on open-sourcing your proposed model?

**Limitations:**

Please include some discussion on negative downstream impacts of the proposed work (e.g., these types of representations are commonly used for voice cloning, so the proposed work may increase the quality of systems used for non-consensual voice cloning).

---

> ### Author Rebuttal · Authors · 2023-08-10
>
> We thank Reviewer RFky for taking the time to review our paper and providing constructive suggestions to improve the paper. Below we answer the question raised in the review.
>
> ---
>
> > Section 4.6 is well done, although the “mapping phones to codewords” section could be made more compelling by utilizing the mapping to perform phoneme classification and producing a phoneme error rate.
>
> Using the mapping in Figure 4 results in a frame-wised phone error rate of 58.2% (done by assigning each codeword to the dominant phone and treating all other phones assigned to the codeword as error). We thank the reviewer for the suggestion and will add this to section 4.6.
>
> ---
>
> > The bolding in table 5 confuses me, both because there are multiple bold items per column, and some columns have no bold items without a clear reason.
>
> We are sorry for the confusion. The goal of Table 5 is to measure how close are the distributions between learned discrete units and phonemes; active cluster and code perplexity are not bolded since they can not reflect the goal. These two metrics are simply here to show the codebook utilization rate. (A larger value does not indicate better performance, nor does a smaller value; e.g., a perfect codebook can have high or low utilization rate as long as each codeword is mapped to no more than one phone.) For metrics that can reflect our goal, namely Cls Pur., Phn Pur., and PNMI, we highlight the best result in online/offline clustering separately. We thank the reviewer for pointing this out and will add more detailed description to the table.
>
> ---
>
> > Previously, open-source releases for this research task (e.g., wav2vec 2.0) have contributed significantly to downstream research efforts. Do you intend on open-sourcing your proposed model?
>
> The source code (as in appendix) will also be published after the anonymous period.
>
> ---
>
> > Please include some discussion on negative downstream impacts of the proposed work (e.g., these types of representations are commonly used for voice cloning, so the proposed work may increase the quality of systems used for non-consensual voice cloning).
>
> Although the method might not be useful for tasks that are not related to phonetic/semantic contents of speech,  we thank the reviewer for the reminder and we will add a paragraph accordingly to list the risk (in different downstreams) in the main paper in the next version.

---

> > ### Comment · Reviewer_RFky · 2023-08-12
> >
> > I thank the authors for their revision. The weaknesses and limitations I mentioned have been addressed in this iteration. The correspondence between the discovered units and known phoneme categories strengthens the authors' argument. Prior open-source work in this domain (e.g., wav2vec 2.0) has drawn significant interest from the research community--as well as citations. This paper has the potential to do the same. I raise my review score from a 6 to a 7.

---

### Official Review · Reviewer_UMJ9 · 2023-07-10

**Soundness:** 4 excellent
**Presentation:** 3 good
**Contribution:** 3 good
**Rating:** 7
**Confidence:** 4

**Summary:**

The paper proposes a self-supervised paradigm combining the methods of self-distillation and online clustering. Specifically for each frame, a teacher model's activations are clustered based on initialized codebooks. The codebooks are then updated using a momentum based method. Then a student model is trained to predict the codebook index for each of masked frames. Authors run extensive experiments on ASR, acoustic unit discovery and SUPERB tasks comparing their methods with SOTA baselines such as wave2vec2, HuBert, data2vec and WavLM. While their method is simple in the sense the codebook vectors are not updated based on gradients, they either beat or perfrom very similar to the strong baselines used.

**Strengths:**

The key strength of the paper is simplicity with which the code-books are updated combining the which is sort of momentum based k-means clustering rather than gradient based updating such as in wave2vec2 framework. At the same time they use a different teacher network embeddings unlike HuBERT model. Combing these two ideas leads to strong state-of-art performance. The cluster analysis results such as cluster purity, codebook perplexity and number of active clusters further show the effectiveness of this method over other online and offline clustering based self-supervised methods. The plots showing P(phone|code) gives additional insight how the phones are clustered to only a few code vectors. the concentration of phones over  that The paper further extends its scope to multilingual experiments. The paper is well-written and enough details are provided for reproduction of experiments.


**Weaknesses:**

1. No explanation given for hyper-parameter tuning experiments especially why we see the huge change as the top N layers for clustering is changed.
2. Why did the authors choose to sum the loss over top N layers instead of just using one layer? Each layer has different codebook associated which leads to an increase in number of code-books to be kept track of and hence more computation.

**Questions:**

1. Did the authors do any study to compare the effect of summing the loss over all frames vs just the masked frames?
2. How does the conditional prob P(phone|codebook) look like as one goes closer to top layers?
3. Table 7 for multilingual experiments in supplementary material is not clear to me. For layer 5, Authors show that 1.1% of code-words are shared across 9 languages but it steeply jumps to 86.5% when all 10 languages are considered. What is the implication of that?
4. Figure 3 caption seems like it should be changed as N refers to number of code-books there.

**Limitations:**

Authors have experimented mainly on English datasets and hence they acknowledge the risks associated with neglecting other languages. But they do provide some results on multilingual experiments showing acoustic units are shared across languages in an attempt to mitigate that concern to a small extent.

---

> ### Author Rebuttal · Authors · 2023-08-10
>
> We thank Reviewer UMJ9 for taking the time to review our paper and recognizing our contribution. Below we answer the question raised in the review.
>
> ---
>
> > No explanation given for hyper-parameter tuning experiments especially why we see the huge change as the top N layers for clustering is changed.
> > Why did the authors choose to sum the loss over top N layers instead of just using one layer? Each layer has different codebook associated which leads to an increase in number of code-books to be kept track of and hence more computation.
>
> As demonstrated in Section 4.5 and Figure 3, the number of codebooks is the most dominant hyper-parameter of the proposed method. As a reference, using only single codebook at the top layer results in a high WER 87.4 (under the same setup mentioned in Section 4.5; evaluate at 200k with fixed LM decoding parameters).
> A key fact is the information encoded by Transformer models can change significantly from layer to layer [A]. Targeting more layers therefore provides richer targets that avoid the model collapsing into layer-specific content. As a result, using more target layers naturally leads to a better model. However, early layers are usually less related to the underlying phonemes[A], thus the performance decreases as we involve more than 10 layers. We thank the reviewer for raising the doubt and will add more explanation to the experiment sections for better clarity.
>
>
> ---
>
> > Did the authors do any study to compare the effect of summing the loss over all frames vs just the masked frames?
>
> We followed the common practice of MLM methods to compute loss in only the masked frames and did not experiment with unmasked loss. As a reference, prior work combining offline clustering and MLM in speech (HuBERT; [20]) discovered computing loss at unmasked position results in worse performance.
>
> ---
>
> > How does the conditional prob P(phone|codebook) look like as one goes closer to top layers?
>
> Please refer to the global rebuttal section where the pdf file is attached. A quick takeaway is that while all layers share a similar structure, latter layers tend to assign more codes to the more frequent phones, leaving some less frequent phones to be under-represented.
> This also supported our explanation on the benefit of targeting more layers.
>
> ---
>
> > Table 7 for multilingual experiments in supplementary material is not clear to me. For layer 5, Authors show that 1.1% of code-words are shared across 9 languages but it steeply jumps to 86.5% when all 10 languages are considered. What is the implication of that?
>
>
> The implication of Table 7 is to show that codewords are modeling fine-grained acoustic units that are language-independent. E.g., the majority (86.5%) of the codewords at layer 5 are shared by all 10 different languages. Even the least general codewords are shared by 3 languages, meaning there are no language-specific codewords learned through training.
>
> ---
>
> > Figure 3 caption seems like it should be changed as N refers to number of code-books there.
>
> The number of codebooks N is the same as the number of layers used for clustering (one codebook each), we will update the caption to make it more consistent.
>
> ---
>
> Finally, we thank the reviewer for pointing out typos in the manuscript that will be fixed in the next version.
>
>
> #### References mentioned in rebuttal
>
> [A] Layer-wise Analysis of a Self-supervised Speech Representation Model, https://arxiv.org/pdf/2107.04734.pdf

---

### Official Review · Reviewer_Z4mQ · 2023-07-10

**Soundness:** 3 good
**Presentation:** 4 excellent
**Contribution:** 3 good
**Rating:** 5
**Confidence:** 4

**Summary:**

The paper introduces a method called DinoSR for improved speech representation learning. DinoSR combines three existing key concepts: masked language modeling, self-distillation and on-line clustering. The authors demonstrate that these components complement each other and lead to a better model for speech representation learning by evaluating the learned representations on downstream tasks.

The DinoSR works as follows. First, contextualized embeddings are extracted from the input audio using a teacher network. Next, an online clustering system is applied to these embeddings, resulting in a machine-discovered phone inventory. This step helps in identifying distinct units or phonemes present in the speech data. Finally, the discretized tokens from the clustering step are used to train the student network using MLM loss. (The teacher network params are EMA of student network).

**Strengths:**

* The paper combines different contemporary approaches for speech representation learning to achieve a better representation learning.
* Removes the offline clustering constraint of HuBert like approaches.

**Weaknesses:**

* Small dataset evaluation
  - The results presented in the paper use a small dataset in the context of speech representation learning. It is not clear if the benefits of the proposed approach are still relevant compared to contemporary approaches in big data regime.

* Generalizability to other downstream tasks
  - The speech representation extracted by the proposed method seem to superior to contemporary approached in ASR/phoneme related tasks. However, it is not clear if they extend to other downstream tasks such as speaker recognition/ emotion recognition etc. (Table 3)

* Generalizability to other sizes or model architectures.
  - The choice of target layers in teacher model is a critical hyper-parameter. It is not clear if they have to be re-tuned for every new architecture or model size for optimal downstream performance.

**Questions:**

* How does the proposed method compare to w2v-bert approach?
* Can the authors add ASR results without LM.

**Limitations:**

None.

---

> ### Author Rebuttal · Authors · 2023-08-10
>
> We thank Reviewer Z4mQ for taking the time to review our paper and expressing explicit concerns. Below we answer the question raised in the review.
>
> ---
>
> > Small dataset evaluation
>
> > Generalizability to other sizes or model architectures
>
> We would like to emphasize that the model size and dataset used in this work, although not gigantic, are still standard setups considered by the recent prior works [9,13,20,22,34]. As mentioned in Section 4.1, the dataset and model size we selected to use is constrained by the computing resource (16 V100 GPUs maxed out our capacity, scaling up to the model size by 2x would require 4x more GPUs according to the prior works). In addition, prior works [9,20] also found scaling is not a problem for these MLM-based Transformers. As a supporting measure, the source code (as in appendix) will also be published after the anonymous period, so the model can easily be experimented at a larger scale.
>
> ---
>
> > Generalizability to other downstream tasks
>
> Similar to ContentVec [34], this work focused on learning phonetic/semantic contents in speech and conducted a diverse set of evaluations toward the goal. Consequently, other contents (such as speaker information) are overlooked and expected to be neglected by the model. We apologize if the goal is not clear enough in the current version. We will update the manuscript to make it more explicit.
>
> ---
>
> > How does the proposed method compare to w2v-bert approach?
>
> From a high-level point of view, the most significant differences between DinoSR and w2v-BERT are as follows:
> - In addition to the MLM objective with discrete tokens, w2v-BERT also includes a contrastive objective similar to wav2vec-2.0.
> - DinoSR built discrete targets from contextualized representation with online learning as described in the paper; w2v-BERT built a discrete target from localized CNN representation using linear projection and the Gumbel softmax activation.
>
> In practice, there are also various differences such as the input surface feature (waveform v.s. spectrogram), basic building block (Transformer v.s. Conformer), etc. Moreover, w2v-BERT was proposed with 24-layer Comformer Encoder and is not publically available, hence it’s hard to compare the two methods apple-to-apple.
>
> ---
>
> > Can the authors add ASR results without LM.
>
> Here we provide the ASR result without LM decoding. Greedy search is used and we also list the prior works [13,22] reporting WER without LM decoding.
>
> | | dev-clean | dev-other | test-clean | test-other |
> |---|---|---|---|---|
> | *10 min fine-tuning* | | | | |
> | wav2vec 2.0 [13] | 46.1| 51.5| 46.9| 50.9 |
> | DinoSR | 33.5| 37.1| 33.9| 37.1 |
> | *1hr fine-tuning* | | | | |
> |wav2vec 2.0| 24.1| 29.6| 24.5| 29.7|
> |WavLM [22] | -| - | 24.5| 29.2|
> |DinoSR| 17.6| 21.8| 17.7| 2.1|
> | *10hr fine-tuning* | | | | |
> |wav2vec 2.0| 10.9| 17.4| 11.1| 17.6|
> |WavLM| - | -| 9.8| 16.0|
> |DinoSR| 7.5| 12.2| 7.7| 12.5|
> |*100hr fine-tuning* | | | | |
> |wav2vec 2.0|6.1|13.5|6.1|13.3|
> |WavLM|-|-|5.7|12.0|
> |DinoSR|4.7|9.8|4.5|9.9|
>
> ---
>
> We thank the reviewer again for raising these questions, we will modify the paper to incorporate these additional information/clarification in the next version. Finally, we hope the reviewer can understand the limitations of our resource and weight the concern on scaling less in the final review.

---

> > ### Comment · Reviewer_Z4mQ · 2023-08-21
> >
> > I thank the authors for their response. Most of the references pointed out ( [9,13,20,22,34]) regarding dataset size, show results for Librilight along with librispeech.  Nonetheless, based on authors overall response, i updated my score.

---

### Official Review · Reviewer_Jtdv · 2023-07-10

**Soundness:** 3 good
**Presentation:** 3 good
**Contribution:** 3 good
**Rating:** 7
**Confidence:** 3

**Summary:**

The paper introduces DinoSR, a method that combines masked language modeling, self-distillation, and online clustering for self-supervised speech representation learning.

The authors demonstrate that these concepts complement each other and result in a strong representation learning model for speech. DinoSR extracts contextualized embeddings using a teacher network and applies an online clustering system to discover meaningful acoustic units. The discretized tokens from the clustering process guide a student network. The paper claims that DinoSR outperforms previous state-of-the-art methods in several downstream tasks and provides a detailed analysis of the model and the learned discrete units.

The key innovation of DinoSR, which is the introduction of a gradient-free online clustering method that leads to meaningful acoustic units. The authors emphasize their contributions in advancing the state-of-the-art in various benchmarks through end-to-end training and providing a closer examination of embeddings from speech transformers via discrete units. They also mention future work possibilities, including structural learning with the codebook, scaling to larger models, and extending the model to different modalities.

**Strengths:**

Novel Approach: DinoSR combines self-distillation, online clustering, and masked language modeling for self-supervised speech representation learning.

Detailed Analysis: The paper provides a detailed analysis of the model and the learned discrete units. This analysis offers insights into the underlying representations and contributes to a deeper understanding of speech processing.

Key Innovation: Introducing a gradient-free online clustering method, leading to meaningful acoustic units, is highlighted as a key innovation of DinoSR. This innovation adds value to the field by providing a method for discovering and utilizing important acoustic units in speech representation learning.

The proposed approach does very well on the general benchmarks showing its usefulness across tasks.

**Weaknesses:**

Limited Discussion of Limitations: The paper does not explicitly discuss the limitations of DinoSR. It is crucial to acknowledge any potential drawbacks or constraints of the proposed approach to provide a balanced view of its strengths and weaknesses.

Also, no ablation based on what performance gains are achieved after each stage is not provided, which would have helped understand the different building blocks of the proposed approach.

**Questions:**

A note on the limitations of the proposed approach would have been useful.

**Limitations:**

The paper does not discuss the limitations of the proposed approach.

---

> ### Author Rebuttal · Authors · 2023-08-10
>
> We thank Reviewer Jtdv for taking the time to review our paper and recognizing our contribution. Below we answer the question raised in the review.
>
> ---
>
> > Limited Discussion of Limitations: The paper does not explicitly discuss the limitations of DinoSR. It is crucial to acknowledge any potential drawbacks or constraints of the proposed approach to provide a balanced view of its strengths and weaknesses.
>
> Limitations are provided in Section A1. However, we agree that more limitations should be discussed in the paper, e.g., the method might not be useful for tasks that are not related to phonetic/semantic contents of speech. We thank the reviewer for the reminder and will add a paragraph accordingly to the main paper in the next version.

---

### Official Review · Reviewer_Maey · 2023-07-25

**Soundness:** 3 good
**Presentation:** 3 good
**Contribution:** 3 good
**Rating:** 7
**Confidence:** 4

**Summary:**

This paper proposes DinoSR, a self-supervised training method with self-distillation and online clustering. The key contribution of this work is the online clustering which is a gradient-free method to learn acoustic unit representations. The authors demonstrate that this approach outperforms previous state-of-the-art models on resource limited automatic speech recognition and the discretizing cluster is closely aligned with the human phonetic. The paper is well-written and provides clear explanations of the proposed approach. The experimental results are extensive and demonstrate the effectiveness of proposed works.

**Strengths:**

1) The article is well structured and details the specific methodology of DinoSR, whose combination of masked language modeling, self-distillation, and online clustering is relatively innovative and can be compared convincingly with the current better self-supervised representations.
2) The work is experimentally more complete and has a high confidence level.
3) The authors provide the code which will benefit the community.

**Weaknesses:**

1) The contributions of this paper are somewhat limited. Self-distillation and MLM task have been used in lots of previous work such as data2vec and SPIRAL. Furthermore, the online clustering has also been used in many previous works such as wav2vec 2.0 and vq-wav2vec.
2) It would be appreciated if the authors could provide more theoretical analysis about why the proposed method works well, especially in alignment with phonetic unit and the high quality of codebook. It will make the effectiveness of the work more persuasive.

**Questions:**

1) As the key contribution of this work is the online clustering, it is suggested to provide some theoretical analysis or experiments to clarify why online clustering can align phonetics effectively.
2) This work does not take methods to keep codebook diversity, such as diversity loss in wav2vec 2.0, but has a very good codebook active rate. It would be valuable to analyze the reason with more explanations or experiments.
3) There are numerous online clustering methods available. Is the capability of model architecture having impact on the codebook quality? For example, VQ-APC still uses RNN, but wav2vec 2.0 and DinoSR use Transformers. Is the current comparison in a fair condition?
4) The following should be further clarified:
* The “contextualized” is mentioned several times. To my knowledge, wav2vec 2.0 also learns the contextualized representations. But the paper states that wav2vec 2.0 is clustering of non-contextualized representations. Why?
* Line 104-105 and equation 2, I am not sure whether equation 2 is accurately defined.
* The demonstration of codebook update in section 3.2 is not sufficiently clear. The initialization of s_v and n_v in equation 3 is not mentioned in either the paper or the appendix. Though I was finally able to find this information in the code, it would be beneficial to include it in the paper.
* Line 238, should the "default 8" for codebook size V be some other digit, e.g. 256?
* Figure 3, the legend of y axis is missing.
* In section 4.6, only the 5th layer is chosen for analysis. Why is this layer selected? What about the other layers? Is the perplexity same for all the layers?

**Limitations:**

The author does not talk about the limits of his article.

---

> ### Author Rebuttal · Authors · 2023-08-10
>
> We thank Reviewer Maey for taking the time to review our paper and providing constructional feedbacks. Below we answer questions raised in the review.
> (quotes from the original review are trimmed to save space)
>
> ---
>
> > The contributions of this paper are somewhat limited...
>
> We would like to highlight our contribution from a different point of view focusing more on the use of discretized representation learning. To the best of our knowledge, existing self-supervised learning methods including vector quantization have all been following the paradigm proposed in VQ-VAE [A] - using discrete representation as information bottleneck in the forward pass. This work is novel in the way we propose to learn discrete units that serve as self-supervised learning targets instead of information bottleneck. More importantly, our tokenizer is jointly learned with the self-supervised learning model itself, pointing out a different path for self-supervised learning methods in different fields [B,C] that have been relying on the quality of pre-trained tokenizers.
>
> ---
>
> > It would be appreciated if the authors could provide more theoretical analysis about why the proposed method works well, ...
>
> > As the key contribution of this work is the online clustering, it is suggested to provide some theoretical analysis or experiments to clarify why online clustering can align phonetics effectively.
>
> > This work does not take methods to keep codebook diversity, ..., but has a very good codebook active rate. It would be valuable to analyze the reason with more explanations or experiments.
>
>
> Let $x$ be the input speech and $z$ be the codewords from the codebooks of the teacher model, we show that the goal of DinoSR is to maximize the lower bound of the mutual information betwee $x$ an $z$ , i.e.,
>
> $I(x;z) = H(z) - H(z|x)$.
>
> - The first term $H(z)$ is the codebook perplexity. While perplexity can be difficult to control for gradient-based VQ methods, the proposed online clustering method can easily achieve high perplexity as shown in Table 5. Our explanation is that DinoSR performs online clustering in an embedding space that changes slowly throughout the training, since the teacher model is an EMA of the student model. To be more specific, our method stands in the middle ground between clustering frozen feature (e.g., upper part of Table 5) and clustering fast-changing features (e.g., prior works in lower part of Table 5), hence it is more robust to code collapse but still falls slightly behind offline clustering as discussed in Section 4.6.
>
> - For the second term $H(z|x)$, our training objective is to minimize the cross entropy between the teacher model $P(z|x)$ and the student model $Q(z|x)$, i.e., minimizing $H(P,Q)$.  From the property of entropy, we know that minimizing $H(P,Q)$ is minimizing the upper bound of $H(P)$, which corresponds to the last term of the mutual information above.
>
> Finally, though there are many different contents in speech that could be captured by the codewords upon training, we hypothesize that acoustic units are particularly more likely to be modeled by our framework than other contents such as speaker information, background sound, etc.  Our reasons are (1) the frame rate of the model is ~50hz, meaning each frame corresponded to only 20ms of the speech; and (2) as mentioned in Section A2, we perform instance normalization on the pre-quantize feature, which can be viewed as removing inter-utterance information prior to clustering.
>
> ---
>
> > ... Is the capability of model architecture having impact on the codebook quality? For example, VQ-APC still uses RNN, but wav2vec 2.0 and DinoSR use Transformers. Is the current comparison in a fair condition?
>
> As the title of  section 4.6 suggested, we aimed to analyze the quality of clusters instead of competing against other methods in this section with Table 5 and Figure 4 & 5. We agree with the concern on the architecture difference between VQ-APC / Co-training APC and other transformer-based methods, but we also think they should not be removed in order to credit these preceding works on online clustering. We thank the review for raising the concern, we will add a caveat to the table to direct the reader to focus more on comparing transformer-based models against each other.
>
> ---
>
> > ... the paper states that wav2vec 2.0 is clustering of non-contextualized representations. Why?
>
> Wav2vec 2.0 clustering comes from the VQ layer that takes the output of CNN over waveform as input, resulting in a fixed receptive field for each token that contains only local information (but the representation of wav2vec 2.0 is indeed contextualized as suggested by the reviewer).
>
> ---
>
> > Line 104-105 and equation 2, I am not sure whether equation 2 is accurately defined.
>
> Apologize for the confusion, L104 contains a typo ($Z^k_t$ should be $Z^k_v$).  Eq. 2 defines a subset of the teacher model representations ($\tilde{z}^t_k$ defined in L86) which will be used to update each codeword $v$. We thank the reviewer for pointing this out and will improve this in the next version.
>
> ---
>
> > The demonstration of codebook update in section 3.2 is not sufficiently clear. ... Though I was finally able to find this information in the code, it would be beneficial to include it in the paper.
>
> We thank the reviewer for pointing this out, $s^k_v$ is initialized to $e^k_v$ (which is random initialized as discussed in Section A.2) and $n^k_v$ is initialized to 1 for all $k$ and $v$. We will add these to the next version.
>
> ---
>
> #### References mentioned in rebuttal
> - [A] Neural Discrete Representation Learning, https://arxiv.org/pdf/1711.00937.pdf
> - [B] HuBERT: Self-Supervised Speech Representation Learning by Masked Prediction of Hidden Units, https://arxiv.org/abs/2106.07447
> - [C] BEiT v2: Masked Image Modeling with Vector-Quantized Visual Tokenizers, https://arxiv.org/pdf/2208.06366.pdf

---

> > ### Comment · Reviewer_Maey · 2023-08-19
> > **Thank the authors for the feedback**
> >
> > I thank the authors for the feedback.  The authors responses have addressed most of my concerns raised in the previous review. I suggest the authors incorporate their responses (especially the theoretical analysis) into the new revision of the paper. I raise my score from 6 to 7.

---

### Official Review · Reviewer_zY5D · 2023-07-26

**Soundness:** 3 good
**Presentation:** 3 good
**Contribution:** 2 fair
**Rating:** 6
**Confidence:** 5

**Summary:**

This paper proposed DinoSR, a novel self-supervised learning (SSL) approach for speech representations that combines the idea of the masked language model, self-distillation, and online clustering. It improves the data2vec method by using online clustering to obtain discrete targets from the teacher model and receives better results in multiple tests.

**Strengths:**

The paper is well-written and easy to follow. A rich set of experiments were conducted, and decent results were obtained: the proposed method sometimes achieved state-of-the-art (SOTA) results and sometimes not too far behind SOTA.

**Weaknesses:**

1. The major weakness to me is originality when compared to data2vec. Although the main idea of the paper is presented as combining the masked language model, self-distillation, and online clustering together, the joint use of the first two was actually studied in data2vec. Thus an alternative interpretation of the novelty is perhaps an improvement of data2vec by using discrete rather than continuous targets from the teacher.

2. Some secondary weaknesses:
A. In Eq. 2, which frames are chosen to generate $\tilde{z}^k_t$?

B. Masked Language Modeling (MLM) -> defined twice.

C. In variables like $\tilde{z}^k_t$, better to quote the superscript $k$ with a bracket (e.g. $\tilde{z}^{(k)}_t$) to avoid confusion.

D. Not sure if the results of the probability-based metrics in Table 5, such as perplexity and mutual information, are comparable for the online clustering systems since different numbers of active clusters are results that lead to incomparable sharpness of distributions.

E. In the reference sections: volume and pages are missing for journal references; "In" is missing in NIPS references, such as [4] and [11].

**Questions:**

"By labelling each codeword using articulation manner classes in English, we revealed the fact that some of the acoustic attributes are embedded in the high-dimensional space. For example, vowels and silences demonstrated a high degree of concentration."

Could you please explain how can this be revealed in a two-dimensional space?
I personally interpret the figure from a reverse perspective: Figure 5 verified the codebooks have good correlations with articulatory features, as many belonging to the same category have a high degree of concentration.

**Limitations:**

1. Though some initial multilingual results were presented in the supplementary materials, in the paper itself, the proposed method is only verified using English data.
2. Only clean data was used in the training and test. Perhaps worth investigating the use of the proposed method on data with general audio events and noises.
3. Since online clustering was performed on different layers, it would interesting to understand further the differences and similarities of the codebooks obtained based on different layers.

---

> ### Author Rebuttal · Authors · 2023-08-10
>
> We thank Reviewer zY5D for taking the time to review our paper and providing the detailed feedback. Below we answer concerns/questions raised in the review.
>
> ---
>
> > The major weakness to me is originality when compared to data2vec. Although the main idea of the paper is presented as combining the masked language model, self-distillation, and online clustering together, the joint use of the first two was actually studied in data2vec. Thus an alternative interpretation of the novelty is perhaps an improvement of data2vec by using discrete rather than continuous targets from the teacher.
>
> We would like to highlight our contribution from a different point of view focusing more on the use of discretized representation learning. To the best of our knowledge, existing self-supervised learning methods including vector quantization have all been following the paradigm proposed in VQ-VAE [A] - using discrete representation as information bottleneck in the forward pass. This work is novel in the way we propose to learn discrete units that serve as self-supervised learning targets instead of information bottleneck. More importantly, our tokenizer is jointly learned with the self-supervised learning model itself, pointing out a different path for self-supervised learning methods in different fields [B,C] that have been relying on the quality of pre-trained tokenizers.
>
> ---
>
> > In Eq. 2, which frames are chosen to generate $\tilde{z}^k_t$?
>
>  $\tilde{z}^k_t$ are from the teacher network (as defined in L85) which takes the unmasked audio as input.
>
> ---
>
> > Not sure if the results of the probability-based metrics in Table 5, such as perplexity and mutual information, are comparable for the online clustering systems since different numbers of active clusters are results that lead to incomparable sharpness of distributions.
>
> The comparison is fair since all methods are given the same amount of freedom (256 clusters). Probability-based metrics are comparable in this case, and the number of active clusters should *not* be considered for these metrics. For example, consider a perfect codebook (i.e., each phoneme is exclusively represented by one codeword and all other codewords are inactive), we will have low number of active clusters (sharp distribution) but high MI.
>
> ---
>
> > Could you please explain how can this be revealed in a two-dimensional space? I personally interpret the figure from a reverse perspective: Figure 5 verified the codebooks have good correlations with articulatory features, as many belonging to the same category have a high degree of concentration.
>
> We are sorry for the confusion, we are trying to point out the fact that the codebooks have good correlations with articulatory features as you mentioned, i.e., the codewords that represent silences are concentrated, and the codewords that encode vowels are also concentrated. We will revise the wording to make it clearer.
>
> ---
>
> Finally, we thank the reviewer for pointing out typos in the manuscript that will be fixed in the next version. We also recognized the limitations pointed out by the reviewer and will seek to cover them in our future work.
>
>
>
> ### References mentioned in rebuttal
> - [A] Neural Discrete Representation Learning, https://arxiv.org/pdf/1711.00937.pdf
> - [B] HuBERT: Self-Supervised Speech Representation Learning by Masked Prediction of Hidden Units, https://arxiv.org/abs/2106.07447
> - [C] BEiT v2: Masked Image Modeling with Vector-Quantized Visual Tokenizers, https://arxiv.org/pdf/2208.06366.pdf

---

### Author Rebuttal · Authors · 2023-08-10

Attached pdf file contains figures that display $P(phone|code)$ for different layers per reviewer UMJ9's request.

---

### Comment · Area_Chair_ev7f · 2023-08-15

Dear reviewers,

The authors have uploaded their rebuttal.  Please take time to go over it.  If you have any further questions or concerns regarding the authors' rebuttal, please start a discussion.   If you are willing to adjust your scores after reading the rebuttal, please do.   For those who have already done it, thanks!

Best,

AC

---

### Decision · Program_Chairs · 2023-09-21

**Decision:**

Accept (poster)

**Comment:**

This paper proposes a so-called DinoSR framework where the authors combine self-distillation and online clustering for self-supervised learning of speech representations and show its effectiveness in extensive experiments in a variety of downstream applications.  While the work is closely related to existing techniques such as masked language modeling, self-distillation and Hubert,  this work learns a tokenizer in a self-distillation framework with a gradient-free online clustering.  The performance is strong where SOTA results are achieved in some of the tasks and others are on-par with SOTA. The authors also provide a detailed analysis of the model and resultant acoustic units.  Overall, the work is interesting and may have its value to the speech machine learning community.   There is a consensus among the reviewers to accept this paper.